# Research on Wafer-Level MEMS Packaging with Through-Glass Vias

**DOI:** 10.3390/mi10010015

**Published:** 2018-12-28

**Authors:** Fan Yang, Guowei Han, Jian Yang, Meng Zhang, Jin Ning, Fuhua Yang, Chaowei Si

**Affiliations:** 1Institute of Semiconductors, Chinese Academy of Sciences, Beijing 100083, China; yangfan3104@semi.ac.cn (F.Y.); hangw1984@semi.ac.cn (G.H.); yangjian@semi.ac.cn (J.Y.); zhangmeng@semi.ac.cn (M.Z.); ningjin@semi.ac.cn (J.N.); 2College of Materials Science and Opto-Electronic Technology, University of Chinese Academy of Sciences, Beijing 100049, China; 3School of Microelectronics, University of Chinese Academy of Sciences, Beijing 100049, China; 4School of Electronic, Electrical and Communication Engineering, University of Chinese Academy of Sciences, Beijing 100049, China; 5State Key Laboratory of Transducer Technology, Chinese Academy of Sciences, Beijing 100083, China

**Keywords:** wafer level packaging, through glass via (TGV), laser drilling, MEMS devices

## Abstract

A MEMS fabrication process with through-glass vias (TGVs) by laser drilling was presented, and reliability concerns about MEMS packaging with TGV, likes debris and via metallization, were overcome. The via drilling process on Pyrex 7740 glasses was studied using a picosecond laser with a wavelength of 532 nm. TGVs were tapered, the minimum inlet diameter of via holes on 300 μm glasses was 90 μm, and the relative outlet diameter is 48 μm. It took about 9 h and 58 min for drilling 4874 via holes on a four-inch wafer. Debris in ablation was collected only on the laser inlet side, and the outlet side was clean enough for bonding. The glass with TGVs was anodically bonded to silicon structures of MEMS sensors for packaging, electron beam evaporated metal was used to cover the bottom, the side, and the surface of via holes for vertical electrical interconnections. The metal was directly contacted to silicon with low contact resistance. A MEMS gyroscope was made in this way, and the getter was used for vacuum maintenance. The vacuum degree maintained under 1 Pa for more than two years. The proposed MEMS fabrication flow with a simple process and low cost is very suitable for mass production in industry.

## 1. Introduction

Wafer-level packaging is a key technology to guarantee the MEMS devices’ lifetime and reliability, and the proportion of the cost is over 30% in the fabrication of MEMS devices. Interconnection and bonding are two main problems in MEMS wafer-level packaging. There are many ways to achieve electrical interconnection, such as through-silicon via (TSV) technology, silicon on insulator (SOI) technology, glass-in-silicon (GIS) technology, and diffusion resistance technology [1,2]. TGV technology is also used for interconnection [3,4]. There are various bonding technologies, such as eutectic bonding, thermal bonding, direct bonding, and anodic bonding.

Interconnections need high electrical properties of conductive materials. Diffusion resistance technology, SOI technology, and GIS technology that use silicon as conductive material is only suitable for low-frequency domains such as sensors. However, TGV and TSV technology, which use metal as a conductive material, can be applied in the radio frequency domain [5]. TSV can realize high density interconnection. Therefore, it is most commonly used in microelectronics and MEMS fields. However, the fabrication process is very complicated [6,7]. TGV has not been widely used at present due to forming holes on glasses [8,9,10]. However, the TGV technique is very attractive due to glass’ superior mechanical, thermal, and chemical resistance, and low dielectric constant [11].

Mechanical drilling, wet drilling, sand blasting, inductively coupled plasma (ICP) etching, and laser drilling are the most widely used methods for forming vias in glasses and there are also some new methods, like discharge methods, glass reflow methods, ultrasonic drilling methods, and wet etching of photo-structural glass [12], such as Foturan and Apex glass [13]. The mechanical drilling has a rough machined surface, and its diameter is limited to 100 μm [14]. Wet drilling cannot make holes with a high aspect ratio due to isotropic etching. Sand blasting can machine thousands of through holes simultaneously at high accuracy, but particles usually stick to the glass surface, which is a problem for wafer bonding [15]. The ICP etching rate is very slow and the depth cannot be very large as a thick mask is difficult to pattern [16]. Cost and efficiency limit the application of glass reflow methods and ultrasonic drilling methods. Discharge methods can obtain high aspect ratios as well as smooth-machined surfaces [17].

Laser drilling does not need a mask, machining speed can be up to 2000 µm/s [18], the via aspect ratios are as high as 70 [19], and it is a good option for MEMS industry. Hwoever, there are still some problems, as bulges around the holes influence further bonding and via metallization [20,21]. Applying the method of liquid-assisted laser processing (LALP) can reduce the temperature gradient and facilitate the debris ejection bulges. However, it will increase instrument complexity [19]. Additionally, the expensive femtosecond laser or excimer laser used for etching glass has low etching rates together with the high cost even though they can achieve a smooth surface [22]. Silex Corporation has studied the TGV’s RF character, and the wafer bonding is metal diffusion or eutectic. The holes are fully filled with metals for interconnections through thick electroplating of gold and copper [23]. By studying the laser drilling technique on glasses, a more reliable and less costly method, anodic bonding with TGVs for packaging is revealed in the paper. Additionally, the interconnection is realized by electron beam deposited metal covering the bottom and the side of the holes to reduce the process time and cost. An accelerometer and a gyroscope are made in this way, in particular, the vacuum sealing chamber of the MEMS gyroscope using the getter had a stable vacuum. Its vacuum degree maintained under 1 Pa for more than two years.

## 2. Materials and Methods

### 2.1. Theory of Laser Drilling

Laser drilling is a technology using a laser as a heat source to produce micro-holes. The laser has good monochromaticity and directionality, as well as high energy. It can transfer photon energy of the laser light to glass, then melt the material and remove the material through vaporization or sublimation, then make a via hole [24,25,26].

When using a laser to drill, firstly, the focal point should be made to scan along a certain line on the surface of the glass. A certain thickness of the glass drilling process can be accomplished. Then it moves the focal point down by a certain displacement and scans along with the same line in the focal plane to finish the etching of the next layer of glass. Etched by scanning in the focal plane after several times, the TGV process can be achieved. The key parameters of laser drilling include the laser power, scanning path, step depth, and step times.

A solid-state picosecond laser, AOPico 532, produced by INNO Laser, Shenzhen, China, was used for the drilling experiment. It has a wavelength of 532 nm, a power of 10 Watts, and a pulse width of seven picoseconds. The frequency is 100 kHz, and the laser spot diameter is less than 10 μm when focused on the glass surface. Since the glass is a transparent material, the absorption ability to visible light is poor. Therefore, we need to improve the laser power in the drilling process [27,28]. In the experiment, the output power must be over 60% to start the drilling process. At this point, the glass began to melt or sublimate after absorbing enough laser energy. We set the laser power to be 95% to study the laser drilling in glass during our process.

To find the minimum via diameter in the drilling research on 300-μm thick glass, circular paths are used. First, we draw some circles with different diameters and let the laser move along the circular path to complete a drilling process with a certain thickness. To guarantee the effect of a single layer, the laser scanned along the single layer and repeated several times. After drilling one layer of glass, then laser focused down to the next layer, completed the drilling process of the next layer, then moved down step by step until getting through the whole glass. The distance of layers is called the step depth, and the layer number is called the step time.

The efficiency of the drilling is dependent on the scan path circle diameter, the scan times, and the step times. For a smaller diameter, fewer scan times and step times means less drilling time. Eight holes with diameters of 40 μm, 45 μm, 50 μm, 55 μm, 60 μm, 70 μm, 80 μm, and 90 μm are drawn in a line with a center distance of 0.5 mm, and the process parameters are gradually reduced to improve the efficiency of the drilling. When the scan times is 4, the step depth was 80 μm, the step time was 12, and the process time was 58 s. However, holes whose scan path diameter is from 45 to 55 μm are irregular circles, and the hole whose scan path diameter is 40 μm gets partl of the way through, as shown in Figure 1a. To improve the hole quality, a reasonable condition was that the scan time is set to 5, the step depth was set to 80 μm, the step time was set to 15, and the process time was 1 min and 29 s. The results are shown in Figure 1.

As can be seen, a hole with a diameter less than 40 μm is difficult to drill through. Due to reflections of energy on the sidewalls, the laser cannot reach the bottom of the hole. Only the scan path circle with a diameter more than 50 μm can be drilled thoroughly. The holes with a path diameter of 50 μm are not very irregular, but acceptable, which can be used in MEMS packaging. Holes with path diameters more than 50 μm can be regular and better.

The hole measured about 88.9 μm in diameter at the front side, and 48.2 μm in diameter on the back side, as shown in Figure 2. By measuring the exit and the entry we can find that the inclination of the sidewall is about 3.89°.

In order to verify the density of drilling, the spacing of holes’ centers was set as 200 μm, and the drilling process was achieved with good effect, as shown in Figure 3. We drew 90 circles with diameters of 50 μm within 20 × 20 mm^2^. Drilled with the optimized process, the drilling time was 7 min and 4 s, and it cost about 9 h and 58 min for drilling 4874 via holes on a four-inch Pyrex 7740 wafer.

Next, an experiment on 500-μm thick Pyrex 7740 glass was made. At first, we used a circular path as the drilling path. However, we could not achieve drilling with a completed hole on the back side, no matter how we adjusted the circular path and the layer thickness. Therefore, we chose the concentric circles as the laser path on each layer. Then, after optimizing the parameters, we found that setting the spacing of the concentric circles to 4–6 μm can ensure efficient glass drilling of each layer. The circular path and concentric circular path are shown in Figure 4. Additionally, the step depth is 80 μm and the step time is 25.

Figure 5 shows the back side and front side of holes with different diameters by drilling along a concentric circular path. Among them, the minimum diameter is 80 μm, and the maximum diameter is 160 μm. It can be concluded that, compared with the circular path, the concentric circular path has better adaptability. We can drill different sizes of holes in glasses with different thicknesses.

Except the step depth, step times, and scan path, laser power, pulse width, and frequency should also be considered. Since the laser power used is limited, a total power 10 W is increased from 60% to 98%, the processability is improved, and 95% of the power is used in the study. In addition, the laser pulse width and frequency were changed to observe the drilling effect, but it seems there as little or no effect. Thus, according to the drilling demands regarding holes of different diameters or in glasses of different thicknesses, the key parameters one should optimize are the step depth, step times, scan path, scan times, and laser power. More reasonable parameters mean higher efficiency.

Figure 6 shows the SEM scanning picture of drilling holes with a path diameter of 80 μm. The hole is measured about 115.1 μm in the diameter at the front side and 48.2 μm on the back side. The dip angle of the sidewall is 3.89°.

### 2.2. Fabrication Process of MEMS Device

In order to verify that laser drilling can be used for wafer-level packaging of MEMS devices, we designed fabrication processes of a MEMS gyroscope and accelerometer based on TGV.

Figure 7 shows the fabrication process flow diagram of the main works in MEMS wafer-level packaging. The process of creating MEMS devices based on TGV is as follows:
(a)The device is made on a SOI wafer. Firstly, the top silicon layer of the SOI wafer is etched by ICP, to form the structure layer;(b)Then the structure layer is released by HF steam;(c)The movable area for the structure layer is etched on the glass;(d)The TGV is made by laser drilling;(e)The anodic bonding of the glass and SOI wafer is completed to achieve the wafer-level packaging;(f)Lastly, the metal is filled by physical vapor deposition (PVD) at the side of the glass with holes after drilling in order to achieve the electrical interconnection. The electrodes are patterned by using spray coating and wet etching processes.

## 3. Results

In order to verify whether the metal can cover the sidewall, we drilled a row of holes in 500-μm thick glass. The glass was used as the hard mask to cover the silicon wafer. Then Ti and Au with thicknesses of 50 nm and 450 nm were deposited in the holes by an electron beam evaporation process. After deposition, we moved the glass away then measured the profile of the metal film on the surface of silicon by 3D microscope, as shown in Figure 8. The diameter of the film is about 50 μm, and the thickness of film is 1.41 μm. Part of the metal was not stuck to the sidewall, but continued to fall to the bottom of the holes.

After drilling, the glass was cleaned by using H_2_SO_4_ and H_2_O_2_, then anodically bonded to a silicon wafer. The result of the bonding is shown in Figure 9. It can be seen that the edge of the hole is smooth, and there are no particles on the bonding surface. The bonding area is very clean and the bonding is good enough for MEMS packaging.

We also tested the conductive ability of the metal in the holes. Glass with holes was bonded with P-type silicon with a resistivity of 0.001–0.005 Ω·cm^−1^. Then we filled the holes with metal Ti and Au by evaporation. We then patterned by spray coating. After removing of photoresist, the ohmic contact was formed by rapid annealing at 450 °C for 30 min. Then we measured that the resistance between the holes at a distance of 500 μm. The measurement results obtained before and after annealing are shown in Figure 10. It can be clearly seen that good ohmic contact was achieved. The resistance was calculated as 800 mΩ from the slope of the voltage-current curve. This proved that the holes were effectively covered by metals and the metals formed good ohmic contact with silicon.

The glass after drilling was used for MEMS device packaging as the above process, then the packaging of a ring-type MEMS gyroscope and the comb micro-accelerometer were achieved, respectively. Figure 11 is the ring-type gyroscope packaged by TGV and Figure 12 shows the comb accelerometer packaged by TGV.

The experiments proved that the holes in glass drilled by laser can meet the requirements of MEMS devices’ electrical interconnections and wafer-level packaging. At the same time, the packaging cost and reliability are better than TSV because of using an anodic bonding process.

In the experiments, when encapsulating the gyroscope, the Ti and Au were sputtered as a getter, then activated at 400 °C for 1.5 h in the bonding process. Through long-term testing, the vacuum sealing chamber of the MEMS gyroscope using the getter has a stable vacuum. Its vacuum degree maintains under 1 Pa for more than two years. There are no significant changes, as shown in Figure 13.

## 4. Conclusions

Laser drilling on glasses studied in the paper provides a good way to fabricate MEMS devices. The minimum diameter of via holes on 300-μm thick glass is only 48 μm, and the via pitch distance can be reduced to 200 μm, which meet most MEMS device packaging demands. A metal film of only hundreds of nanometers is capable of covering the side of the via holes and realizing the interconnection between the silicon structure and the outer space. A ring gyroscope was packaged with glasses with TGVs, and the vacuum level remained unchanged under 1 Pa. The packaging reliability is also proved.

Additionally, the proposed MEMS fabrication method with TGV has a simple process, which does not require complex equipment and process conditions, and it is suitable for the development of laboratory prototype devices and industrial mass production.

## Figures and Tables

**Figure 1 micromachines-10-00015-f001:**
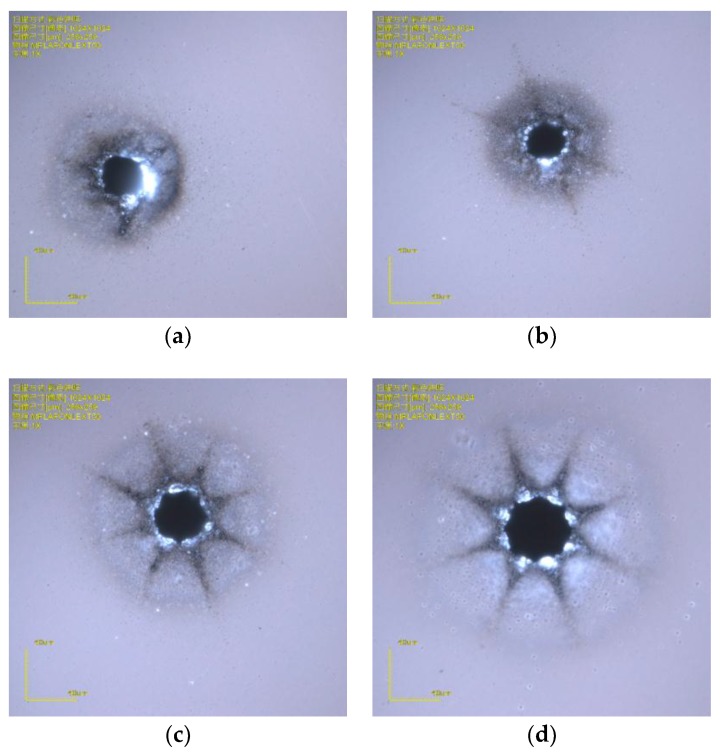
Via holes with different laser path diameters (**a**) 40-μm path diameter with 37.9-μm exit; (**b**) 50-μm path diameter with 49.6-μm exit; **(c**) 60-μm path diameter with 58.1-μm exit; and (**d**) 70-μm diameter with 67.7-μm exit.

**Figure 2 micromachines-10-00015-f002:**
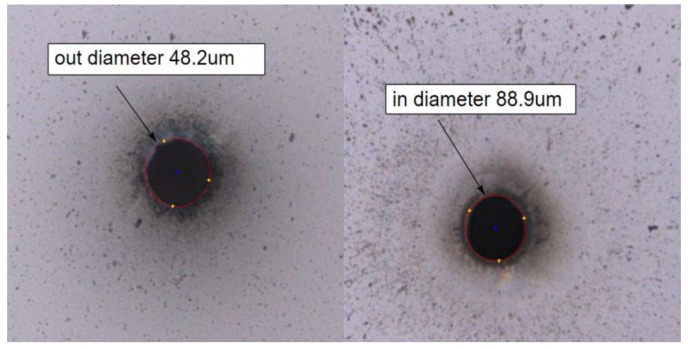
Picture of a via hole drilling using optimized parameters.

**Figure 3 micromachines-10-00015-f003:**
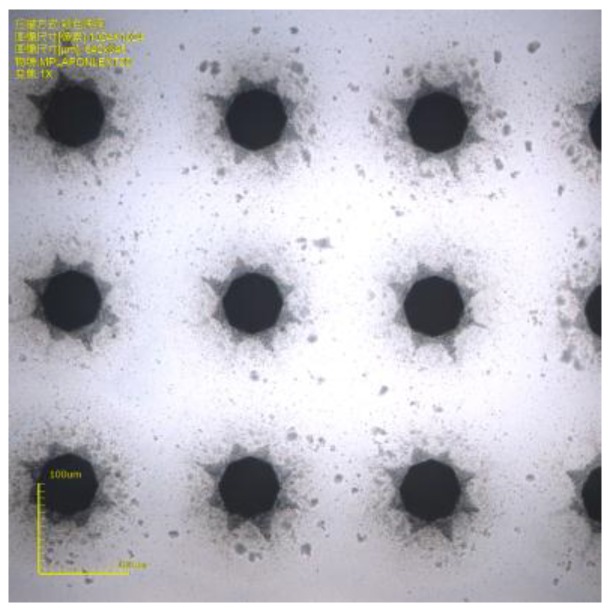
Picture of via holes with a 200 μm pitch distance.

**Figure 4 micromachines-10-00015-f004:**
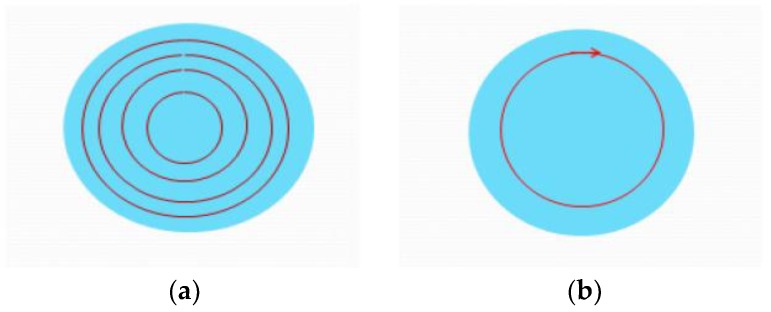
Schematic of laser paths: (**a**) Concentric circles; and (**b**) circle.

**Figure 5 micromachines-10-00015-f005:**
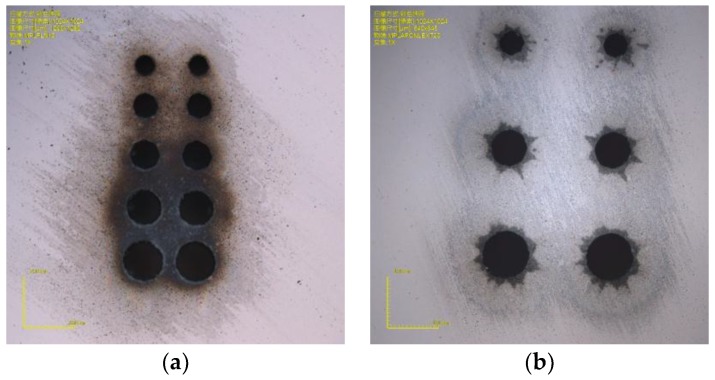
Pictures of via holes on 500-μm Pyrex 7740: (**a**) inlet surface; and (**b**) outlet surface.

**Figure 6 micromachines-10-00015-f006:**
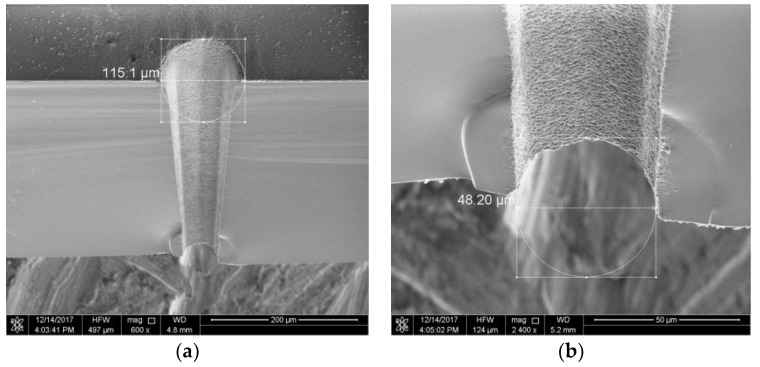
SEM cross-section picture of a via hole on 500-μm Pyrex 7740: (**a**) inlet; and (**b**) outlet.

**Figure 7 micromachines-10-00015-f007:**
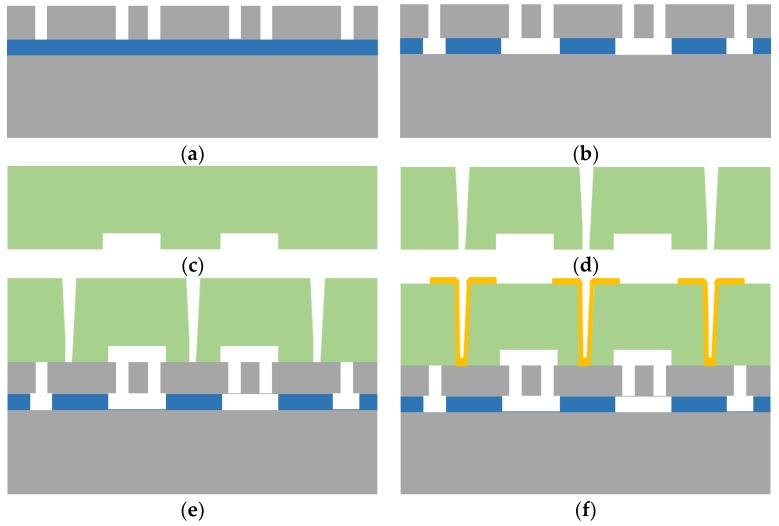
Fabrication process flow of a MEMS device with TGV: (**a**) Structure etch; (**b**) HF release; (**c**) space etch; (**d**) TGV drilling; (**e**) anodic bonding; and (**f**) patterned metal deposition.

**Figure 8 micromachines-10-00015-f008:**
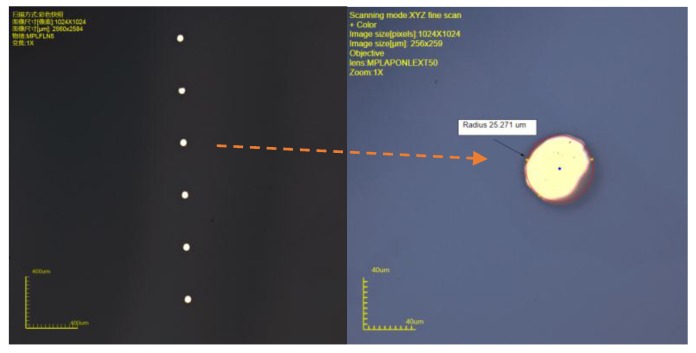
3D scanning picture of metal film.

**Figure 9 micromachines-10-00015-f009:**
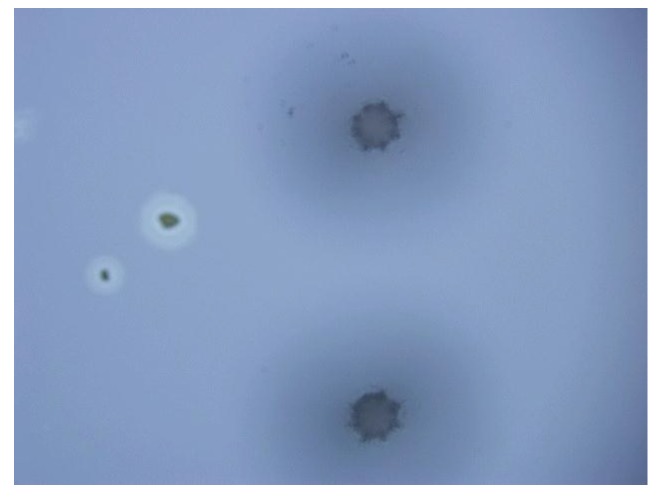
Picture of the bonding area of via holes and silicon.

**Figure 10 micromachines-10-00015-f010:**
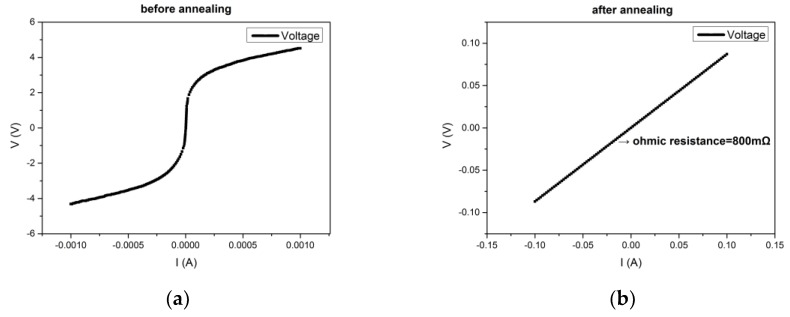
The voltage-current testing results using B1500A (**a**) before annealing, and (**b**) after annealing.

**Figure 11 micromachines-10-00015-f011:**
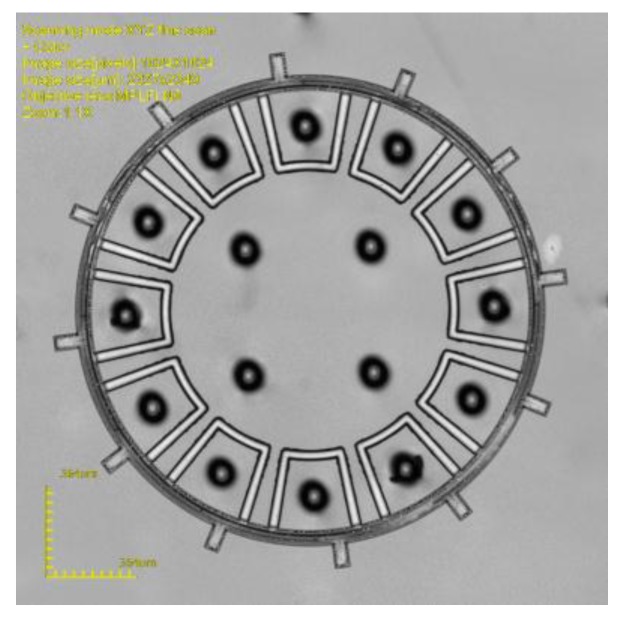
Picture of a ring gyroscope.

**Figure 12 micromachines-10-00015-f012:**
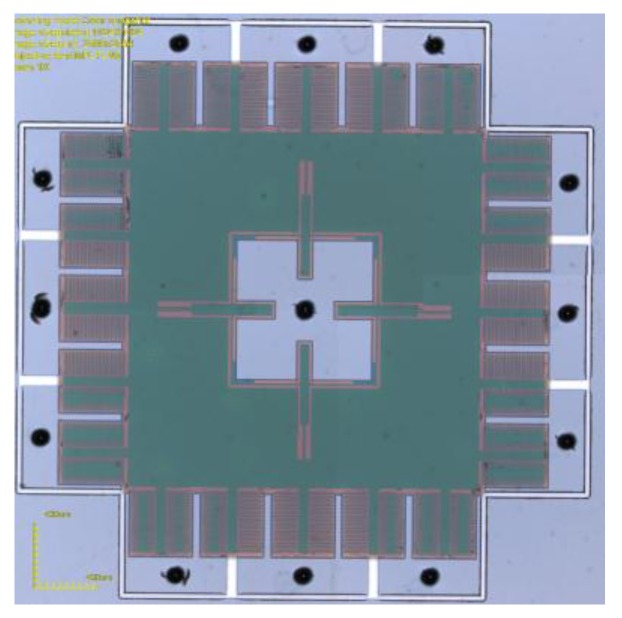
Picture of an accelerometer.

**Figure 13 micromachines-10-00015-f013:**
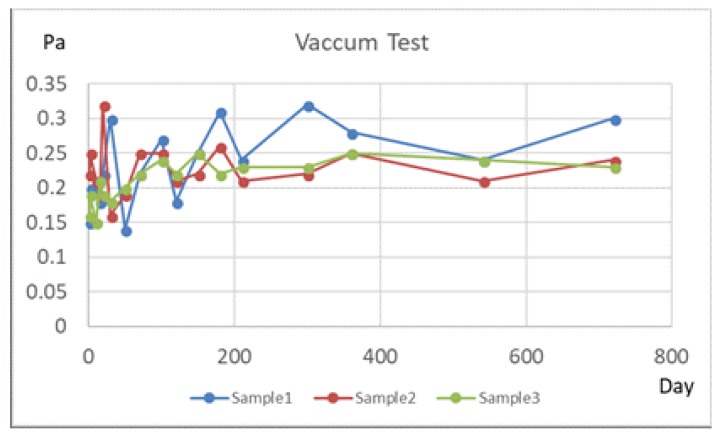
Vacuum tests of the MEMS cavity packaged by glass with TGV and a getter.

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
