# Peer review of "Research on Wafer-Level MEMS Packaging with Through-Glass Vias"

_micromachines, 2018, doi:10.3390/mi10010015_

Round 1
Reviewer 1 Report
This research paper describes the development of a laser drilling process in glass substrates for MEMS device packaging. The study describes the effect of laser path on the resulted through-glass vias. This article also shows examples of the packaging of MEMS inertial sensors using this technology. Following are some comments based on the current manuscript:
1. This manuscript must highlight its novelty. The laser drilling process is not new, this manuscript should emphasize its contribution to this area.
2. Line 14, how is the fabrication efficiency improved? Quantitative results are suggested.
3. Line 43, “it has a fast speed and high efficiency”. This statement is not always true, depending on the fabrication volume.
4. The introduction section should include more literature review on existing laser drilling technologies, and highlight this article’s novelty and contribution.
5. Line 64, the model number of the laser drilling machine must be included. In addition, what is the diameter of the focal point?
6. Line 77, please confirm the equipment setting of step depth, 10 nm step depth seems not practical. In addition, if step time is 50 with a 10 nm step depth, the final penetration depth is only 0.5um. Please double check.
7. In Figure 1, what is the cause of the prominent patterns? Why this pattern disappears in Figure 2, and appears again in Figure 3?
8. Line 90-91, 50um step depth * 15 steps = 750um, which exceeds the thickness of the substrate. Is it feasible? Same for line 109.
9. Line 93, the inclination angle should be close 4 deg. How is the 0.068 deg. calculated? Same for line 132. The angle is calculated incorrectly.
10. Figure 6, images with better qualities should be used.
11. Line 137, what it the thickness for Ti and Au, respectively? How is the evaporation done? Any shadow effect in such a high aspect ratio structure.
12. Line 145, “The bonding area is very clean…”, but in Figure 8, particles at the bonding interface are obvious. Figure 8 also requires a scale bar.
13. Line 154, Ohmic contact is determined by the shape of IV curve, rather than using the resistance value.
14. Section 2.1, laser drilling step depth and step number are the two parameters that have been studied. What are the important conclusions based on the study? If a glass substrate with a different thickness would be drilled, how could other researcher develop/optimize their processes based on this study? Also, other critical parameters, such as laser power, pulse width, etc., are not studied/optimized, which makes this study incomplete.
Author Response
Thanks a lot for your suggestions, sir. Here are what I`ve revised about the paper, besides, there are also some parts revised under other reviewers.
1. This manuscript must highlight its novelty. The laser drilling process is not new, this manuscript should emphasize its contribution to this area.
Thanks for the reviewer’s suggestion. The novelty is highlight in abstract and introduction now, in particular to line12: reliability concerns about MEMS packaging with TGVs, likes debris and via metallization, were overcome.
2. Line 14, how is the fabrication efficiency improved? Quantitative results are suggested.
Thanks for the reviewer’s suggestion. This part of work is added in line 106 to line 112, the minimum drilling time is dependent on the quality of via holes.
3. Line 43, “it has a fast speed and high efficiency”. This statement is not always true, depending on the fabrication volume.
It’s very good a question, and the sentence is deleted from the paper.
4. The introduction section should include more literature review on existing laser drilling technologies, and highlight this article’s novelty and contribution.
Thank you for the reviewer’s suggestion. Glass drilling technologies including san blast, wet etching, ICP etching and the likes are reviewed in the third paragraph of Introduction part. More literature review on existing laser drilling technologies are also added in the forth paragraph of introduction part. And the novelty and contribution is highlight in this part:by studying the laser drilling technique on glasses, more reliable and less cost method,anodic bonding for packaging was realized in the paper. And the interconnection is realized by electron beam deposited metal covering the bottom and the side of holes.
5. Line 64, the model number of the laser drilling machine must be included. In addition, what is the diameter of the focal point?
Thanks for the reviewer’s suggestion. The model number is AOPico 532 and the focal point diameter is less than 10μm, the part is added in the third paragraph of Materials and Methods part.
6. Line 77, please confirm the equipment setting of step depth, 10 nm step depth seems not practical. In addition, if step time is 50 with a 10 nm step depth, the final penetration depth is only 0.5μm. Please double check.
I’m very sorry for the mistake, the unit is mm, 0.01mm, which means that the step depth is 10μm initially. Thanks for pointing out.
7. In Figure 1, what is the cause of the prominent patterns? Why this pattern disappears in Figure 2, and appears again in Figure 3?
The prominent patterns looks like radiation shape, it is caused by the force and heat in laser drilling, more drilling time, and the radiation shape is larger, if the drilling time is controlled well in 300μm thickness glass , the radiation shape can be very small, but still exited, as can be seen in gyroscope and accelerometer, the radiation shape is not very clear. But for 500μm thickness glass, no matter how to adjust the drilling conditions, only if the via hole is throughout, the radiation shape is very large. We have wet etched the hole in HF, and the part of radiation shape is etched more quickly than other parts. maybe laser with different wavelength can avoid the problem, so the question is not discussed in the paper.
8. Line 90-91, 50um step depth * 15 steps = 750um, which exceeds the thickness of the substrate. Is it feasible? Same for line 109.
Yes, 50μm step depth is feasible. I’m not very clear about the physical process of laser and glass, but there is a phenomenon revealed in the paper, 500μm thick glass cannot be drilled throughout if laser scan path is circle, and concentric circles path is adopted. I think the explanation is reasonable, enough laser energy for machining can reach the machining surface only if the laser is not absorbed, reflected or refracted too much, but if, the laser direction should be adjusted to make sure there is enough energy on the machining surface. I’m not good at optics, maybe the explanation is wrong, so there is no analysis in the paper, only the result is presented.
9. Line 93, the inclination angle should be close 4 deg. How is the 0.068 deg. calculated? Same for line 132. The angle is calculated incorrectly.
Thanks for pointing out. I’m very sorry for the mistake. The angle is calculated as arctanθ=(88.9-48.2)/2/300=0.068, θ=3.89°. It is corrected in the paper.
10. Figure 6, images with better qualities should be used.
Thanks for the advice, a clearer image is in place.
11. Line 137, what it the thickness for Ti and Au, respectively? How is the evaporation done? Any shadow effect in such a high aspect ratio structure.
Ti and Au is 50nm/450nm, shadow effect is very serious, sputtered metal with an angle of 5° cannot reach the bottom, so electron beam evaporation was used for poor metal particle direction.
12. Line 145, “The bonding area is very clean…”, but in Figure 8, particles at the bonding interface are obvious. Figure 8 also requires a scale bar.
Thanks for the reviewer’s suggestion. I need to say, the picture with not perfect bonding quality is presented intentionally, without the particles, one cannot tell if there is a silicon bonded under the glass with holes, and the particle is not caused by laser drilling, it has a distance from the holes. You can see the gyroscope and the accelerometer picture, there is few particles, and the bonding is nearly perfect.
13. Line 154, Ohmic contact is determined by the shape of IV curve, rather than using the resistance value.
Thanks for the advice, IV curve of figure 9 is added in revised paper.
14. Section 2.1, laser drilling step depth and step number are the two parameters that have been studied. What are the important conclusions based on the study? If a glass substrate with a different thickness would be drilled, how could other researcher develop/optimize their processes based on this study? Also, other critical parameters, such as laser power, pulse width, etc., are not studied/optimized, which makes this study incomplete.
Thanks for the reviewer’s suggestion. Laser power is very important, more power means more machining part at in scan process, the maximum power of used Lasers AOPico 532 is only 10W, the larger the power is, the more powerful the machining ability is. step depth, step times and scan path, scan times, and laser power are very important. Laser pulse width and frequency are also tested, but there is no obvious effect, this part of work is added in last second paragraph of Theory of Laser Drilling section.

Reviewer 2 Report
"Abstract" and "Conclusions" lacks quantitative results, like achieved minimum diameter of TGV and the basic results of vacuum testing.
"Introduction" lacks overwiev of state of the art and how this work aims to expand state of the art. In fact, this work seems not to expand state of the art. See for instance: https://www.google.com/url?sa=t&rct=j&q=&esrc=s&source=web&cd=6&cad=rja&uact=8&ved=2ahUKEwio0IHe4fneAhULkCwKHT8rBtUQFjAFegQIARAB&url=https%3A%2F%2Fieeexplore.ieee.org%2Fiel7%2F6573226%2F6575533%2F06575594.pdf&usg=AOvVaw0xm3v0Xxs5xkwFBs848oK_ even though this paper also has shortcomings in describing state of the art.
In "Materials and methods" shows basically the results of the work, which should be moved to the "Results" chapter.
"References" lacks sufficient papers describing the state of the art and should be substantially improved.
All in all, major revisions are needed; it might be that the systematic evaluation of the different TGV diameters can make a good paper.
Author Response
Thanks a lot for your suggestions, sir. Here are what I`ve revised about the paper, besides, there are also some parts revised under other reviewers.
1 "Abstract" and "Conclusions" lacks quantitative results, like achieved minimum diameter of TGV and the basic results of vacuum testing.
Thanks for the advice, the quantitative results are added in "Abstract" and "Conclusions".
2 "Introduction" lacks overwiev of state of the art and how this work aims to expand state of the art. In fact, this work seems not to expand state of the art. See for instance: https://www.google.com/url?sa=t&rct=j&q=&esrc=s&source=web&cd=6&cad=rja&uact=8&ved=2ahUKEwio0IHe4fneAhULkCwKHT8rBtUQFjAFegQIARAB&url=https%3A%2F%2Fieeexplore.ieee.org%2Fiel7%2F6573226%2F6575533%2F06575594.pdf&usg=AOvVaw0xm3v0Xxs5xkwFBs848oK_ even though this paper also has shortcomings in describing state of the art.
Thanks for the reviewer’s suggestion.State of the art drilling on glasses is added in "Introduction". Compared with other TGV bonding and electrical interconnection way in MEMS, anodic bonding for packaging was realized in the paper, which is seen as the most reliable and lowest cost bonding, and the detailed description is added in the last paragraph in "Introduction".
Thanks for the paper, discharge drilling seems very good a method for drilling, I read some papers, but I don’t know how to realize it, and how long it is on the market. Laser drilling is very mature technology, with my small effort, it is accepted in MEMS industry, I think this is helpful.
3 In "Materials and methods" shows basically the results of the work, which should be moved to the "Results" chapter.
Thanks for the advice, Figure 7 and 8, the electrical interconnection and bonding results is moved to "Results" chapter.
4 "References" lacks sufficient papers describing the state of the art and should be substantially improved.
Thanks for the advice, they are added in revised paper.
5 All in all, major revisions are needed; it might be that the systematic evaluation of the different TGV diameters can make a good paper.
Thanks for the advice, I will try evaluating the factor leading to different TGV diameters in next work.

Round 2
Reviewer 1 Report
1. The novelty of this manuscript is still not clear. Why the reliability concerns with TGV is addressed? Why the debris removing and metallization process is novel? From the manuscript, the reader cannot see any support to these novelties.
2. The prominent pattern shown in Figure 1. The authors thinks they are formed by laser damage and is proved by the faster etching in these areas in HF. Is this defect going to cause any reliability issue? Since this manuscript is to provide a method to fabricate reliable TGVs for MEMS, this issue should be solved. If solved, or the formation mechanism of these defects is studied, the value of this manuscript can be improved a lot.
3. English must be improved.
Author Response
1. The novelty of this manuscript is still not clear. Why the reliability concerns with TGV is addressed? Why the debris removing and metallization process is novel? From the manuscript, the reader cannot see any support to these novelties.
Thanks for your advice, I appreciate your doubts about the novelty, here are the explanations.
(1) “reliability concerns with laser drilling TGV” are shown in reference 19-23, and the problems are summarized and supplemented in the last paragraph of introduction. Reliability concerns in this paper,including debris and via metallization.
(a). debris: Traditional glass drilling using laser processing in air would produce many kinds of defects such as bulges, debris, cracks and scorch. This will influence the fabrication quality for application. Applied the method of liquid-assisted laser processing (LALP) can reduce the temperature gradient, bulges and heat affected zone (HAZ) region for achieving crack-free glass machined holes facilitated the debris ejection. But it will increase instrument complexity. Besides, the expensive femtosecond laser or excimer laser used for etching glass had low etching rates together with the high cost, especially for deep through-wafer etching.
(b) metallization process: Silex cooperation has studied the TGV’s RF character, and the wafer bonding is metal diffusion or eutectic, and the holes are full filled with metals for interconnections through thick electroplating of gold and copper. Electroplating process is commonly used for interconnection.
(2) debris removing and metallization process are not novel, but are problems when using laser drilling glass for MEMS package. In our manuscript, debris collects on the laser entry side, and the exit side is clean and used for bonding, the method avoid bonding problem caused by bugles and debris around the hole, and the bonding is reliable which is verified in MEMS fabrication. That`s novel, and not reported before. Besides, compared with electroplating process, we achieved interconnections by Electron-Beam evaporated after drilling to reduce the process time and the cost.
Our work would provide instructive information for the development of the industrial application of picosecond pulse laser micro-machining of glass for TGV technology and wafer level packaging.
I`m sorry for writing skills, and I hope the explanations are helpful.
2. The prominent pattern shown in Figure 1. The authors thinks they are formed by laser damage and is proved by the faster etching in these areas in HF. Is this defect going to cause any reliability issue? Since this manuscript is to provide a method to fabricate reliable TGVs for MEMS, this issue should be solved. If solved, or the formation mechanism of these defects is studied, the value of this manuscript can be improved a lot.
The advice is very helpful for reliability, according to experiment, prominent pattern is proved no influence to the hermeticity, but whether mechanical shock or stress can damage the glass, that was not tested. I will do some research on that phenomenon next year. And for 300 um thick glass the prominent pattern area can be reduced to a minimum by adjusting the drilling conditions, as shown in figure2, and the reliability influenced by laser damage can be kept to a minimum.
3. English must be improved.
Thanks for the advice, the manuscript was modified in details. I`m trying to improve my English skills.

Reviewer 2 Report
The paper draft is now good with needed modifications made.
In abstract: "It cost about 9 hours and 58
16 minutes for drilling 4874 via holes on a 4-inch wafer." Replace "cost" with "takes"
Author Response
The paper draft is now good with needed modifications made.
In abstract: "It cost about 9 hours and 58 minutes for drilling 4874 via holes on a 4-inch wafer." Replace "cost" with "takes"
Thanks for the reviewer's comments and advices.
It has been modified in the paper: "It took about 9 hours and 58 minutes for drilling 4874 via holes on a 4-inch wafer."

Round 3
Reviewer 1 Report
Thank you authors, the manuscript can be published.